

# Validation and field application of a low-cost device to measure CO₂ and ET fluxes

Reena Macagga[1], Michael Asante[2,3], Geoffroy Sossa[2,4], Danica Antonijević[1], Maren Dubbert[1], Mathias Hoffmann[1]

[1] Leibniz Center for Agricultural Landscape Research (ZALF), Isotope Biogeochemistry and Gas Fluxes, 15374, Müncheberg, Germany

[2] West African Science Service Centre on Climate Change and Adapted Land Use, University of Sciences, Techniques and Technologies of Bamako (USTTB), BP E 423, Bamako, Mali

[3] Council for Scientific and Industrial Research-Savannah Agricultural Research Institute (CSIR-SARI), 00233, Tamale, Ghana

[4] Laboratory of Hydraulic and Water Control, National Institute of Water, University of Abomey-Calavi, Abomey-Calavi, 01 BP 526 Cotonou, Benin

*Correspondence to*: Reena Macagga (Reena.Macagga@zalf.de)



**Abstract**

Mitigating the global climate crisis and its consequences, such as more frequent and severe droughts, is one of the major challenges for future agriculture. Therefore, identifying land use systems and management practices that reduce greenhouse gas emissions (GHG) and promote water use efficiency (WUE) is crucial. This however, requires accurate and precise measurements of carbon dioxide ($CO_2$) fluxes and evapotranspiration (ET). Despite that, commercial systems to measure $CO_2$ and ET fluxes are expensive and thus, often exclude research in ecosystems within the Global South. This is especially true for research and data of agroecosystems in these areas, which are to date still widely underrepresented. Here, we present a newly developed, low-cost, non-dispersive infrared (NDIR)-based, $CO_2$ and ET flux measurement device (~200 Euro) that provides reliable, accurate and precise $CO_2$ and ET flux measurements in conjunction with manual closed chambers. To validate the system, laboratory and field validation experiments were performed, testing multiple different low-cost sensors. We demonstrate that the system delivers accurate and precise $CO_2$ and ET flux measurements using the K30 FR NDIR ($CO_2$) and SHT31 (RH) sensor. An additional field trial application demonstrated its longer-term stability (> 3 months) and ability to obtain valid net ecosystem C balances (NECB) and WUE. This was the case, even though environmental conditions at the field trial application site in Sub-Saharan Africa were rather challenging (e.g., extremely high temperatures, humidity and intense rainfall). Consequently, the developed low-cost CO2 and ET flux measurement device not only provides reasonable results but might also help to democratize science and close current data gaps.

**1 Introduction**

The global climate crisis is one of the most critical problems of our time and identifying and implementing measures to mitigate or adapt to its consequences, such as more frequent and severe drought, is a key challenge. Solving this challenge, requires first and foremost a substantial reduction of anthropogenic greenhouse gas (GHG) emissions in all sectors (IPCC, 2019). While agriculture is a significant contributor to these anthropogenic GHG emissions (FAO, 2020), it might also offer the potential to mitigate the climate crisis by increasing soil carbon (C) sequestration (Lal et al., 2004). Specifically, land use systems and management practices which not only promote a net C uptake but also an efficient water use are needed. They might help to increase soil C stocks and crop productivity, reducing GHG emissions while simultaneously sustaining yield, despite intensifying climate stressors, such as more frequent and severe droughts. Hence, it is crucial to evaluate land use systems regarding their potential to sequester additional C and effectively utilize water. Common parameters used to assess both, are the net ecosystem C balance (NECB; Smith et al., 2010), and the agronomic and ecosystem water use efficiency (WUE; Beer et al., 2009). Their determination, however, requires accurate and precise measurement of carbon dioxide ($CO_2$) and evapotranspiration (ET) fluxes (Chapin et al., 2006; Livingston and Hutchinson, 1995; Rosenstock et al., 2016; Xu et al., 2019).





Measurement of $CO_2$ and ET fluxes are commonly performed using eddy covariance or chamber based systems (Baldocchi et
al., 1996; Smith et al., 2010; Wang et al., 2017; Yang et al., 2014), while especially the latter are well suited for direct treatment
comparisons (Dubbert et al., 2014; Hoffmann et al., 2018; Kübert et al., 2019). In case of a remote study site location or
limitations in power supply, particularly manual closed chamber measurements are used to measure the $CO_2$ exchange and ET
fluxes (Rochette and Hutchinson, 2015). However, the relatively high costs of needed measurement equipment (particularly
gas analyzers) strongly limits their accessibility and often exclude research in ecosystems within the Global South. This
resulted in a pronounced underrepresentation of regions, land use systems and management practices from subtropical and
tropical South America, South Asia, and Africa, even though the quantification of e.g., $CO_2$ fluxes in these regions might
reduce disparities in the global $CO_2$ budget (Canadell et al., 2011; Gurney et al., 2002; Kondo et al., 2015).

Recent efforts to solve this financial constraint focus on developing low-cost, yet reliable, measurement devices. This was
catalyzed by the growing availability of relatively inexpensive microcontrollers, which are increasingly utilized for scientific,
environmental research (Blackstock et al., 2019; Capri et al., 2021). An additional contribution came from the improvement
in accuracy and precision of low-cost relative humidity (RH) and especially non-dispersive infrared (NDIR) $CO_2$ sensors.
Evaluation of commercially-available NDIR $CO_2$ sensors (Keimel et al., 2019; Martin et al., 2017; Pandey et al., 2007; Yasuda
et al., 2012) showed that they have acceptable precision and accuracy in measuring $CO_2$ concentrations especially when proper
calibration methods are applied. Although low-cost NDIR $CO_2$ sensors are commonly used in air quality monitoring studies
(Araujo et al., 2020; Wastine et al., 2022), these sensors have also been applied in environmental research (Bastviken et al.,
2015; Brown et al., 2020). For example, multiple studies have demonstrated the applicability of using low-cost NDIR $CO_2$
sensors for reliable measurements of soil $CO_2$ efflux (Brändle and Kunert, 2019; Curcoll et al., 2022; Harmon et al., 2015) and
water crop use determination (Capri et al., 2021). However, in case of RH sensors, the inversely increased measurement
uncertainty of total water vapor concentration with decreasing RH (e.g. a typical low-cost RH sensor has a measurement
accuracy of 1-3 % in relative but not absolute humidity) might constitute a problem. Despite first studies showing the potential
of using low-cost sensors as an alternative to more expensive commercial counterparts, there is still little evidence that in situ
closed chamber $CO_2$ and ET flux measurements using both, are comparable in precision and accuracy.

Here, we present the hard- and software implementation, as well as laboratory and in situ validation of a newly, low-cost and
open-source $CO_2$ and ET flux measurement device. We hypothesise that by using the device in conjunction with a manual
closed chamber 1.) $CO_2$ and ET fluxes can be reliably and accurately measured; and that 2.) measured $CO_2$ and ET fluxes can
be used to obtain valid estimates of net ecosystem C balance (NECB) and WUE, even under challenging environmental
conditions such as extremely high air temperatures, humidity, and precipitation. To test these hypotheses, we first validated
the accuracy and precision of four different low-cost NDIR $CO_2$ sensors (K30 FR, SCD30, MHZ-14, and MHZ-19) under
controlled laboratory conditions. Afterwards, the NDIR sensors passing laboratory validation as well as two different RH



sensors were validated in field. During field validation, ET and $CO_2$ fluxes (ecosystem respiration ($R_{eco}$) and net ecosystem
exchange (NEE)), as well as temperature-dependent $R_{eco}$ and photosynthetic active radiation (PAR)-dependent gross primary
production (GPP) parameters, were compared to the results obtained simultaneously with a reference infrared gas analyser
(IRGA; LI-850, LI-COR, USA). Finally, the ability of the developed low-cost $CO_2$ and ET flux measurement device to obtain
reliable NECB and WUE as well as its practicability and stability were tested. Therefore, multiple devices were used during a
field trial application in Northern Ghana to obtain seasonal $CO_2$ exchange and ET, as well as NECB and WUE for four different
fertilizer treatments in a maize cultivation.
**2 Material and Methods**
**2.1 Hard- and software implementation**
The developed, highly portable $CO_2$ and ET flux measurement device consists of a logger and sensor unit, both assembled out
of a combination of various low-cost, off-the-shelf components. A complete list of used components, distributors and prices is
given in Table 1. Figure 1 shows the assembled logger and attachable sensor unit, together with a schematic representation of
the wiring. The logger unit consists of an Arduino Uno like microcontroller (Atmega328, AZ-Delivery Vertriebs GmbH,
Germany) with attached Logger Shield module (AZ-Delivery Vertriebs GmbH, Germany) including an SD card reader and
SD card (2 GB) to store sensor readings and a real time clock (RTC) which helps to keep the time and date even when the
system is switched off. A BME280 air temperature (±1 ºC), air humidity (±3 %) and air pressure sensor (±1 hPa; Reichelt
electronics GmbH, Germany) as well as an LCD display (AZ-Delivery Vertriebs GmbH, Germany) and HC-05 Bluetooth
module are part of the logger unit and connected to the microcontroller. The logger unit is fitted into a weather and shock
resistant outdoor housing (B&W Outdoor Case Type 500, OVERHAUL MEDIA GmbH, Germany). The external sensor unit
consists of a NDIR-based $CO_2$ (0-10000 ppm, ±30 ppm ±3 % accuracy; K30 FR, Senseair AB, Sweden), an air humidity (RH)
and air temperature sensor (SHT31, ±2 % accuracy, Sensirion AG, Switzerland or DHT22, ±2 to 5 % accuracy, Aosong
Electronics Co., Ltd, China). Both sensors were connected through a seven core cable to the logger unit using UART (K30
FR) and I2C (SHT31) data communication, respectively. The power supply of the microcontroller is ensured by six
rechargeable AA NiMH batteries (1.2 V; 2600 mAh) in a 6×AA battery holder, which supply 7.2 V. Due to the power
requirements of the external sensor unit (K30 FR and SHT31), an additional 6×AA battery holder is attached to the housing
directly. Software implementation was done using Arduino IDE 2.0.3.








**Table 1:** Sensor components and cost (in Euro) at the time of writing, including weather and shock-proof housing and energy supply (rechargeable batteries). Components needed for optional semi-automatic mode are listed in addition.

| COMPONENT | AMOUNT | DESCRIPTION | PRICE | DISTRIBUTOR |
|---|---|---|---|---|
| B&W OUTDOOR CASE TYP 500 | 1 | Outdoor case for housing electrical components | 28.75 Euro | www.profikoffer.de |
| PVC HARD FOAM PLATE | 1 | PVC 5 mm hard foam plate to create interior of housing for electronic components | 1.5 Euro | www.amazon.de |
| LUSTER TERMINALS | 12 | Luster terminals for wiring electrical components within housing | 0.6 Euro | www.amazon.de |
| 0.2 mm² 24 AWG ELECTRICAL WIRE | | Electrical wires for wiring electrical components within housing | | www.amazon.de |
| 7 PIN AVIATION CONNECTOR | 2 | Aviation connector to connect logger unit within weatherproof housing with passive NDIR sensor installed in the closed chamber to be attached | 2.9 Euro | www.amazon.de |
| 7 CORE RUBBER CABLE (1.5 m) | 1 | Cable to connect logger unit within weatherproof housing with passive NDIR sensor installed in the closed chamber to be attached | 3.75 Euro | www.conrad.de |
| WS R13-112 AAAA ROCKER SWITCH | 1 | Rocker switch for switching on and off | 1 Euro | www.reichelt.de |
| ATMEGA 328 | 1 | Arduino Uno like microcontroller | 5 Euro | www.az-delievery.de |
| DATALOGGER MODULE | 1 | Logger shield for Arduino UNO like microcontroller with SD card reader and RTC unit | 4.6 Euro | www.az-delievery.de |
| HAMA CLASS 4, SD MEMORY CARD, 2 GB, 10 MB/s | 1 | SD memory card to save sensor readings | 6 Euro | www.saturn.de |
| HC-05 BLUETOOTH WIRELESS RF-TRANSCEIVER-MODULE RS232 | 1 | Bluetooth module for wireless communication | 5.2 Euro | www.az-delievery.de |
| 16×2 LCD OR OLED DISPLAY WITH I2C ADAPTER | 1 | LCD or OLED display for data visualization | 3.7 Euro | www.az-delievery.de |
| BMP280 | 1 | Air pressure, air humidity and air temperature sensor | 1.7 Euro | www.reichelt.de |
| DHT22 OR SHT31 MODUL | 1 | Air temperature and air humidity sensor | 6.4 Euro | www.az-delievery.de |
| SENSEAIR K30 FR (FAST RESPONSE) | 1 | $CO_2$ measuring module with fast response time; Measuring range: 0 to 5000 ppm $CO_2$, operating range: 0 to 50 °C | 85 Euro | www.driessen-kern.de |
| GOOBAY 11467 6× (4×) MIGNON (AA) BATTERY HOLDER | 2 (1) | Battery holder for 6× NiMH rechargeable mignon (AA) batteries | 4.6 Euro | www.conrad.de |
| CONRAD ENERGY HR06 MIGNON (AA)-AKKU NiMH 2600 mAh 1.2 V | 12 (16) | NiMH rechargeable mignon (AA) batteries | 38 Euro | www.conrad.de |
| 4.5 V METAL BRUSH AIR PUMP | 2 | Air pump for flushing headspace of small chambers | 9.45 Euro | www.berrybase.de |
| IRLZ44N MOSFET | 1 | Mosfet to control power supply to pumps | 0.75 Euro | www.reichelt.de |
| **TOTAL COST** | | | **199.7 Euro** | |



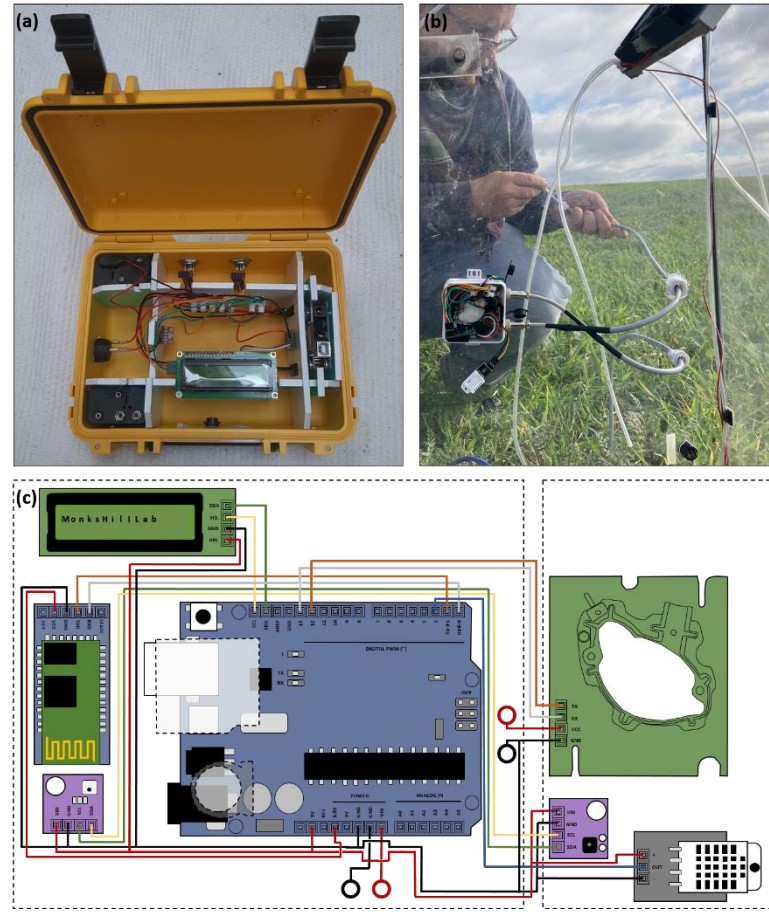


**Figure 1:** (a) Logger unit in weather and shock resistant housing, (b) external sensor unit attached to a transparent non-flow-through non-steady-state (NFT-NSS) closed chamber and (c) schematic representation of wiring.

## 2.2 Laboratory validation

To identify the NDIR sensor most suitable for in situ, dynamic closed chamber measurements, four different NDIR-based sensors, namely 1.) MHZ-19 (Winsen Electronics Technology CO., LTD, China), 2.) MHZ-14 (Winsen Electronics Technology CO., LTD, China), 3.) SCD30 (Sensirion AG, Switzerland) and 4.) K30 FR (Senseair AB, Sweden) were tested and validated regarding their precision and accuracy during a laboratory validation experiment. For this, sensors were placed separately into a sealed, ventilated, cylindrical vessel (Fig. 2; V: 1425.5 cm$^2$) and connected to the developed low-cost logger system.



123

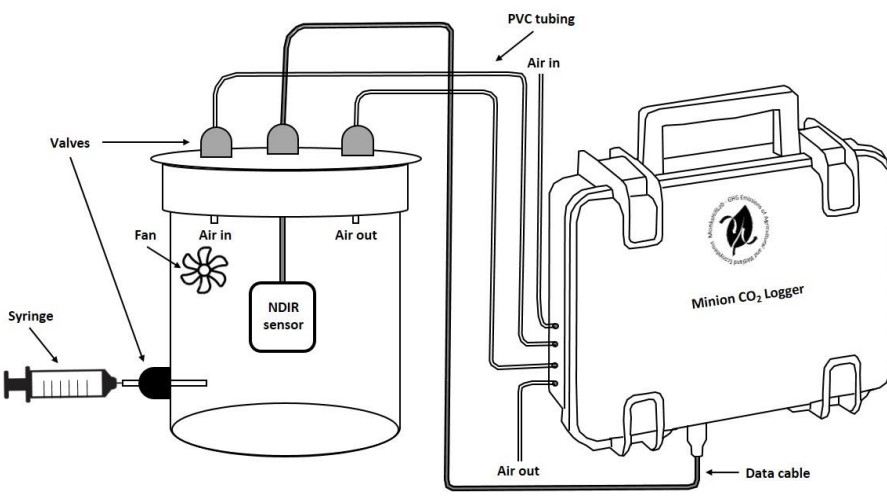

124

**Figure 2:** Experimental setup of the performed laboratory validation experiment for four different NDIR $CO_2$ sensors connected to the developed low-cost $CO_2$ and ET flux measurement device (MHZ-14, MHZ-19, SCD30 and K30 FR). Validation was performed through injecting distinct amounts of technical gas (Linde, Germany; 10000 ppm $CO_2$) into the air-tight, sealed, cylindrical vessel.

All sensors were calibrated in ambient air prior to use according to manufacturer instructions. Afterwards different distinct amounts (5 to 30 ml; in 5 ml steps; each step repeated five times) of a technical gas containing 10000 ppm $CO_2$ (Linde, Germany) were injected into the sealed vessel using a syringe. In between injections, the vessel was flushed with ambient air by two pumps (1.5 L min$^{-1}$) connected to the vessel (semi-automatic measurement mode of the developed device). Finally, $CO_2$ concentration increases inside the vessel, measured in a 5 s interval by the NDIR-based sensors, from before to after injection ($\Delta CO_2$ in ppm) were compared against mixing-induced $CO_2$ concentration increases. Sensors that performed best in terms of accuracy and precision were subsequently validated during the field validation experiment.

**2.3 Field validation**

Field validation of the low-cost $CO_2$ and ET flux measurement device was performed through parallel manual closed chamber measurements using an infrared gas analyzer (IRGA; LI-850, LI-COR, USA) and NDIR sensors ($CO_2$) passing previous laboratory validation, as well as two different RH sensors (ET). Measurements were conducted at the "PatchCrop" experimental field, managed by the Leibniz Centre for Agricultural Landscape Research (Fig. 3; ZALF). "PatchCrop" features multiple smaller patches (72 x 72 m), with diverse and site-specific crop rotations, aiming to create synergies and interactions between fields.



143

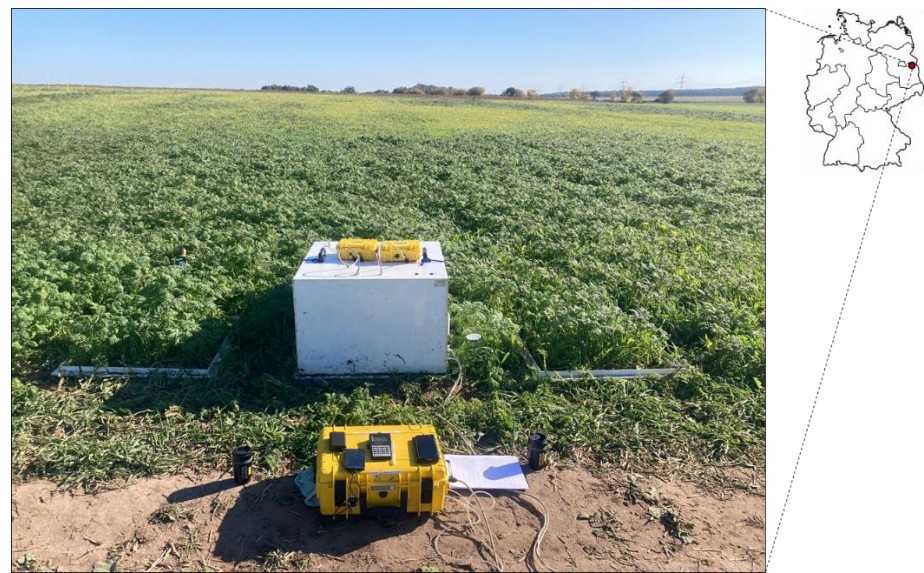

**Figure 3:** Parallel opaque ($R_{eco}$) manual closed chamber measurements with a Li-COR 850 IRGA (LI-850, LI-COR, USA) and the developed, low-cost $CO_2$ and ET flux measurement device at ZALF experimental field near the village of Tempelberg, North-East Germany (52°44'86.2'' N, 14°14'05.1'' E). The developed system was equipped with a K30 FR and SCD30 NDIR, as well as SHT31 and DHT22 sensor.

The experimental field "PatchCrop" is located near the village of Tempelberg, Northeast Germany (52°44'86.2'' N, 14°14'05.1'' E). The temperate climate is characterized by a mean annual air temperature of 9.7°C and mean annual precipitation of 544 mm (ZALF weather station, 2010-2019). The medium loamy, sand textured soil can be classified as Luvisol (WRB). $CO_2$ exchange (NEE and $R_{eco}$) and ET measurements were conducted for a mixture of *Phacelia* and *Guizotia abyssinica* at three repetitive plots, established at one of the patches through installing PVC collars (A: 0.5625 $m^2$; 5 cm deep) in the beginning of October 2022. Measurements started shortly after sunrise and lasted to late afternoon during two consecutive days, using a dynamic, (non-)flow-through non-steady-state ((N)FT-NSS) manual closed chamber system. Used transparent (86 % light transmission; NEE flux measurements) and opaque ($R_{eco}$ flux measurements), cubic shaped PVC chambers had a total volume of 0.296 $m^3$ and were equipped with a fan for efficient headspace mixing. $CO_2$ and $H_2O$ concentrations, as well as RH, during chamber deployment were recorded in parallel using a LI-850 IRGA and the developed, low-cost measurement device, equipped with a K30 FR, SCD30, SHT31 and DHT22 sensor, respectively. NEE, $R_{eco}$, and ET fluxes were measured by alternately deploying the opaque and transparent chambers on the three pre-installed PVC frames. During individual 4 min measurements, $CO_2$ and $H_2O$ concentration, as well as RH, changes in the chamber headspace, air temperature inside and outside the chamber, soil temperature and humidity (TMS-4, TOMST, Czech Republic) as well as PAR (outside the chamber; Skye, UK) were recorded at a 3 s (LI-850) and 5 s interval (NDIR and RH sensors). To validate the low-





cost $CO_2$ and ET flux measurement device, measured $R_{eco}$, NEE and ET fluxes, as well as derived temperature ($R_{eco}$) and PAR
dependency functions (GPP), were directly compared against results obtained in parallel with the LI-850.

**2.4 Field trial application**

The developed, low-cost measurement device has been tested for applicability and reliability under challenging environmental
conditions in an experimental field managed by the Council for Scientific and Industrial Research-Savanna Agricultural
Research Institute (Fig. 4; CSIR-SARI). The experimental field (21 × 54 m), located near the city of Nyankpala, Northern
Ghana (9°24'15.9'' N, 01°00'12.1'' W), featured a split-plot design (3 × 6 m; n=3) with the main plot assigned to tillage
practice (conventional vs. reduced tillage) and the subplot assigned to a factorial combination of organic and mineral fertilizers.
The tropical region around Nyankpala is characterized by a mean annual air temperature of 26 °C and a unimodal rainfall
pattern with a distinct rainy season from June to October followed by a dry season from November to May (Alua et al., 2018)
resulting in a mean annual precipitation of 1100 mm (CSIR-SARI weather station, 1995-2013). The soil is sandy loam textured
and classified as Acrisol (WRB). $CO_2$ exchange (NEE and $R_{eco}$) and ET measurements were conducted for maize (*Zea mays*)
from July to October 2022 at four out of the nine treatments with reduced tillage (bullock plough), namely: 1.) Fertisoil (5 t
ha$^{-1}$; commercial organic fertilizer in Northern Ghana; FT), 2.) farmyard manure (5 t ha$^{-1}$; FM), 3.) Fertisoil + NPK (5 t ha$^{-1}$ +
90-60-60 kg ha$^{-1}$; FT+MIN) and 4.) farmyard manure + NPK (5 t ha$^{-1}$ + 90-60-60 kg ha$^{-1}$; FM+MIN). Measurement campaigns
took place every two weeks from sunrise to late evening using a dynamic, NFT-NSS manual closed chamber system. Used
transparent (86 % light transmission; NEE flux measurements) and opaque ($R_{eco}$ flux measurements), cubic shaped PVC
chambers had a total volume of 1.56 m$^3$ and were equipped with a fan for efficient headspace mixing. $CO_2$ concentration and
RH changes during chamber deployment were recorded using the developed, low-cost measurement device, equipped with a
K30 FR and DHT22 sensor. During each measurement campaign, NEE, $R_{eco}$, and ET fluxes were measured by alternately
deploying the opaque and transparent chambers on pre-installed frames (A: 0.96 m$^2$) at each of the measured plots.



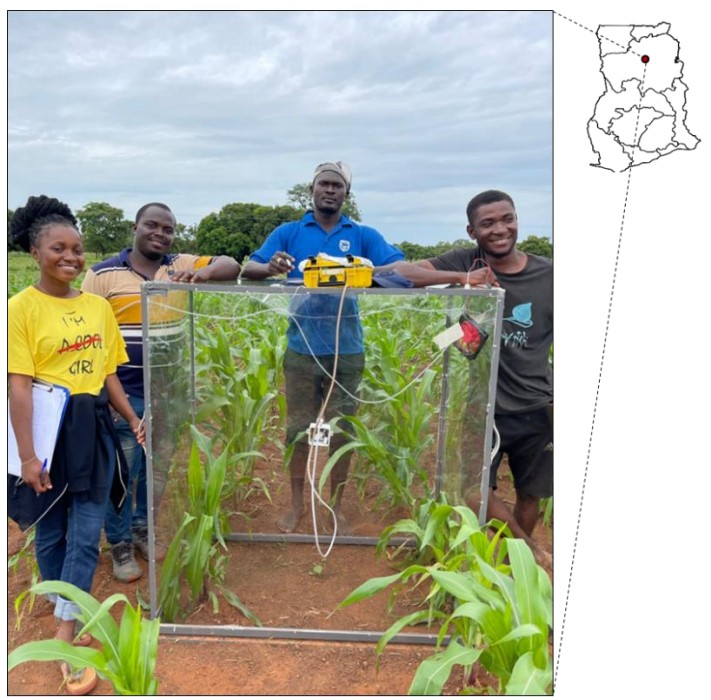

**Figure 4:** Transparent (NEE) manual closed chamber measurement at CSIR-SARI experimental field, used for field trial application of the developed, low-cost $CO_2$ and ET flux measurement device, near the city of Nyankpala, Northern Ghana (9°24'15.9'' N, 01°00'12.1'' W).

**2.5 Data processing**

**2.5.1 $CO_2$ and ET flux calculation, separation and gap-filling**

For laboratory validation, the changes in $CO_2$ concentrations in the vessel, expressed as $\Delta CO_2$ in ppm, were calculated as the mixing ratio of measured ambient air and injected technical gas $CO_2$ concentration (10000 ppm). These were compared with the $\Delta CO_2$ obtained for the four different NDIR sensors as the difference in mean $CO_2$ concentrations measured for one minute right before and two minutes after injection. For the field validation, measured $CO_2$ and ET fluxes were calculated using a modular R script, described in detail by Hoffmann et al. (2015; $CO_2$) and Dahlmann et al. (2022; ET), respectively. Prior to $CO_2$ and ET flux calculation a death-band of 10 % was applied to the data of each chamber measurement. $CO_2$ concentrations measured using the LI-850 were additionally corrected for changes in water vapour during chamber measurements (Webb et al., 1980; McDermitt et al., 1993). Unlike the LI-850 which provided $H_2O$ as mole fraction, used low-cost RH sensors (DHT22 and SHT31) required additional post processing. RH measurements were converted into a mass concentration following Hamel et al. (2015; Eq. 1):





$$H_2O = \frac{RH \cdot e^s}{100 \cdot P}$$                                    (1)

where RH is the relative humidity, P is the gas pressure (Pa) and $e^s$ is the saturated vapour pressure (Pa), calculated according
to Allen et al. (1998). Thereafter, $CO_2$ and ET fluxes were calculated based on the ideal gas law using a linear regression
approach (Eq. 2):

$$f = \frac{MpV}{RTA} \cdot \frac{\Delta c}{\Delta t}$$                                    (2)

where M denotes the molar mass of the gas (g $mol^{-1}$), p denotes the ambient air pressure (Pa) and V denotes the chamber
volume ($m^3$). Since plants accounted for < 0.1 % of the total chamber volume, a static chamber volume was assumed. R denotes
the gas constant (8.314 $m^3$ Pa $K^{-1}$ $mol^{-1}$), T denotes temperature inside the chamber (K), A denotes the basal area ($m^{-2}$) and
$\Delta c/\Delta t$ denotes the linear $CO_2$ (e.g., Leiber-Sauheitl et al., 2014) and $H_2O$ concentration change over time (e.g., Dahlmann et
al., 2022). The variables T and, more importantly, $\Delta c/\Delta t$, were obtained by applying a variable moving window (0.5 to 3 min)
to each chamber measurement. Thus, resulting multiple ET and $CO_2$ fluxes per measurement (based on generated variable
moving window data subsets) were further evaluated according to the following criteria: 1.) fulfilled prerequisites for applying
a linear regression (normality (Lilliefor´s adaption of the Kolmogorov-Smirnov test), homoscedasticity (Breusch-Pagan test)
and linearity); 2.) regression slope ($p \leq 0.1$, t-test); 3.) range of within-chamber air temperature not larger than ± 1.5 K and a
PAR deviation (only transparent chamber measurements) not larger than ± 20 % of the average to ensure stable environmental
conditions within the chamber throughout the respective measurement window; 4.) no outliers present (±6xIQR). Calculated
$CO_2$ and ET fluxes that did not meet all criteria were discarded. In cases where more than one flux per measurement met all
criteria, the $CO_2$ and ET flux with steepest slope and closest to chamber deployment were chosen. For field validation and field
trial application $CO_2$ fluxes were additionally separated into its flux components $R_{eco}$, GPP and NEE and gap-filled through
deriving empirical models. In the case of $R_{eco}$, a temperature-dependent Arrhenius-type function was used and fitted for air as
well as soil temperatures measured in different depths (Lloyd and Taylor, 1994; Eq. 3).

$$R_{eco} = R_{ref} \cdot e^{E_0 \left[ \frac{1}{T_{ref} - T_0} - \frac{1}{T - T_0} \right]}$$                                    (3)

where $R_{ref}$ is the respiration rate at the reference temperature ($T_{ref}$; 283.15 K), $E_0$ is an activation energy-like parameter, $T_0$ is
the starting temperature constant (227.13 K) and T is the mean air or soil temperature during the flux measurement. Out of the
four obtained $R_{eco}$ models (one model for air temperature inside the chamber, one for air temperature outside the chamber; soil
temperature at 2 and 5 cm depth), the model with the lowest Akaike information criterion (AIC) was finally used. In case of
GPP a PAR dependent, rectangular hyperbolic light-response function, based on the Michaelis–Menten kinetic, was used



(Elsgaard et al., 2012; Hoffmann et al., 2015; Wang et al., 2013; Eq. 4). Since GPP cannot be measured directly, GPP fluxes
were calculated as the difference between measured NEE and modelled $R_{eco}$ fluxes, using campaign specific, previously
derived parameters $R_{ref}$ and $T_0$.

$\quad\text{GPP} = \frac{\text{GP}_{max} \cdot \alpha \cdot \text{PAR}}{\alpha \cdot \text{PAR} + \text{GP}_{max}}$ (4)

where $\text{GP}_{max}$ is the maximum rate of C fixation at infinite PAR (µmol $CO_2$ m$^{-2}$ s$^{-1}$), $\alpha$ is the light use efficiency (µmol $CO_2$
µmol$^{-1}$ photons) and PAR is the photon flux density (corrected for chamber light transmission) of the photosynthetically active
radiation (µmol$^{-1}$ photons m$^{-2}$ s$^{-1}$). In cases where the rectangular hyperbolic light-response function did not result in
significant parameter estimates, a non-rectangular hyperbolic light-response function was used (Gilmanov et al., 2007; 2013).
$R_{eco}$ and GPP parameter sets were evaluated and discarded in case of non-significant parameter estimates. If no fit or a non-
significant fit was achieved, averaged flux rates were used for $R_{eco}$ and GPP instead. $R_{eco}$, GPP and NEE were modelled in half
hourly steps for the entire period based on continuously monitored temperature and PAR. For ET, campaign-wise average
daily ET fluxes (for nighttime ET fluxes measured before, for daytime ET fluxes measured after 8:00) were determined and
linearly interpolated between campaigns for the entire crop growth period.
**2.5.2 NECB and WUE**
NECB for the field trial application experiment was calculated as the sum of cumulated NEE, C output such as harvested
biomass C and C input due to organic fertilizer application (Eq. 5; Smith et al., 2010).

$\quad\text{NECB} = \text{NEE} + \text{C}_{input} - \text{C}_{output}$ (5)

Several minor NECB components have not been considered, such as, C input from seeding and methane emissions. However,
due to their relatively low magnitude (e.g., no methane emissions in mineral soil under aerobe conditions) their influence on
the NECB of our study is neglectable. Values for $R_{eco}$, GPP, NEE, harvested biomass C and NECB are given using the
atmospheric sign convention (Ceschia et al., 2010), where positive values indicate C losses from the plant-soil system and
negative values indicate C uptake. Thus, NECB refers to the total change in below-ground C. WUE was calculated as the
agricultural WUE ($\text{WUE}_{agro}$; Eq. 6; Hatfield and Dold, 2019).

$\quad\text{WUE} = \frac{\text{DM}}{\text{ET}}$ (6)

where DM denotes harvested dry biomass in g m$^{-2}$ and ET is cumulative evapotranspiration in mm.





**2.5.3 Error calculation and statistical analysis**

To test for normal distribution of the data obtained from laboratory and field validation measurements, Kolmogorov-Smirnov test ($p<0.05$) was performed. In case of normal distribution, significant differences between $\Delta CO_2$ in ppm or $R_{eco}$, NEE, and ET fluxes measured from low-cost sensors and mixing ratio $\Delta CO_2$ or IRGA-based $R_{eco}$, NEE, and ET fluxes were determined using one-sample t-test ($p<0.05$). Error calculation for $CO_2$ and ET fluxes, as well as crop season $CO_2$ exchange and ET, were quantified using a comprehensive error prediction algorithm described in detail by Hoffmann et al. (2015).

**3 Results and Discussion**

**3.1 Laboratory validation**

Differences in accuracy and precision among the tested, four different low-cost NDIR sensors are shown in Fig. 5a-d as 1:1-agreement plots between mixing ratio (calculated) and measured $\Delta CO_2$. While accuracy can be assessed as deviation from the 1:1-agreement line, precision is determined by the residual standard deviation (SD) and the coefficient of determination ($r^2$) of the linear regression fitted on calculated versus measured $\Delta CO_2$. The K30 FR (Fig. 5d) showed the highest accuracy among all tested NDIR sensors, reflecting well the increase in $CO_2$ concentration ($\Delta CO_2$) derived through mixing ratio. Correspondingly, no significant difference (one sample t-test, $p=0.80$) was found between calculated and measured $\Delta CO_2$. The SCD30 (Fig. 5c), even though fairly accurate at lower, failed to reflect higher calculated $\Delta CO_2$ values and generally tends to overestimate triggered $\Delta CO_2$. Neither the MHZ-14 (Fig. 5b) nor the MHZ-19 (Fig. 5a) were sufficiently accurate and able to reflect triggered $\Delta CO_2$. While the MHZ-14 showed a rather constant offset from the 1:1-agreement by 28 ppm, the MHZ-19 tends to increasingly overestimate higher $\Delta CO_2$ values derived through mixing ratio. Hence, unlike the K30 FR, all other NDIR sensors measured significantly higher $\Delta CO_2$ when compared to mixing ratio $\Delta CO_2$ (one sample t-test, $p<0.01$). Unlike the accuracy, overall precision and measurement repeatability among all four NDIR sensors was generally high and fairly comparable, showing a residual SD of 2.78 ppm, 4.23 ppm, 2.52 ppm and 3.58 ppm, respectively. Regarding the response time (defined as mean time from injection to measured initial $CO_2$ concentration increase), all four NDIR sensors differed substantially, with only 44 seconds for the K30 FR and more than 280 seconds for the MHZ-14. The same was true for the response strength (defined as the mean time from beginning to end of the injection triggered $CO_2$ concentration increase, which represents its steepness), with 61, 160 and 265 seconds for the K30 FR, SCD30 and MHZ-19 respectively. In case of the MHZ-14, response strength could not be evaluated, since no clear saturation after injection induced $CO_2$ concentration increase could be observed.

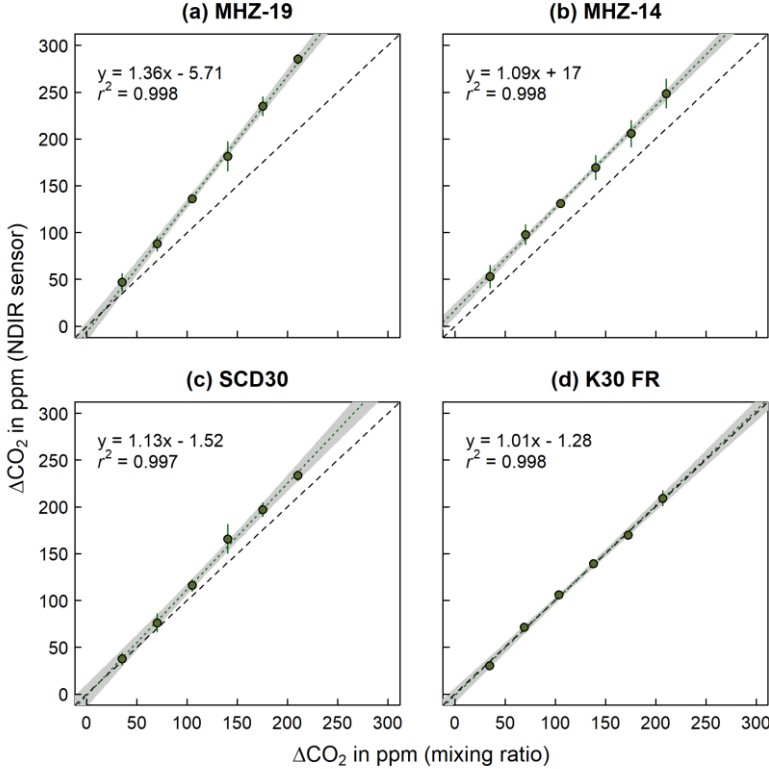

**Figure 5:** 1:1-agreement between mixing ratio and measured $\Delta CO_2$ in ppm from the four low-cost sensors tested (K30 FR, SCD30, MHZ-14 and MHZ-19). The dashed black line indicates the 1:1-agreement. The dotted green line shows the linear regression through the average $\Delta CO_2$ for each injection step (n=5), calculated from the repetitive measurements per step. Error bars indicate ±1.96 SD. The grey shaded area represents the respective confidence band of the regression line.

While accuracy and precision are of course highly relevant, response time and response strength in particular play a key role in determining the extent to which the tested NDIR sensors can be used for in situ NFT-NSS closed chamber measurements. With a response time of almost 2 min and 5 min, respectively, as well as low response strength, MHZ-19 and MHZ-14 would likely fail to correctly reflect $\Delta CO_2$ during short-time (<4 min) closed chamber measurements, regardless of their low accuracy, which makes them additionally unsuitable. Therefore, only the K30 FR (and to a much lower extent the SCD30) with its fast response time and high response strength passed laboratory validation and met all necessary requirements for accurate and precise in situ measurements of $CO_2$ exchange. Our findings, comparing accuracy and precision of four different NDIR sensors during a laboratory setup, are in a good agreement with previous studies performing laboratory validation of single sensors.



Brändle and Kunert (2019), who compared the MHZ-14A NDIR sensor against a GFS-3000 (Heinz Walz GmbH, Germany)
during a laboratory validation observed a similar response time and a general measurement offset of approx. +40 ppm. Based
on this and an additionally conducted field validation, Brändle and Kunert (2019) also suggested that the MHZ-14A is not
suitable for short term measurements (<5 mins). Also findings of González Rivero et al. (2023), who tested the ability of the
SCD30 to reflect calibration gas concentrations and concluded an acceptable accuracy and response time, are in a good
agreement with results of the present study. The most widely tested NDIR sensors so far, however, are those of the K-Series
(as e.g., Ali et al., 2016; Blackstock et al., 2019; Brown et al., 2020; Mendes et al., 2015). Laboratory validation performed by
Blackstock et al. (2019) using K30 1 % sensor to measure a span of different $CO_2$ concentrations verified that it well reflects
$CO_2$ concentrations within the accuracy stated by the manufacturer. Similarly, laboratory tests performed by Mendes et al.
(2015) found that the K30 sensor has nearly perfect linear response against calibration gas $CO_2$ concentrations. Lastly, the
laboratory experiment by Ali et al. (2016) also highlighted the accuracy of the K30 1 % sensor when compared against
measurements of an SBA-5 $CO_2$ gas analyzer (PP Systems, USA). During their experiment both sensors showed a strong
correlation and no offset, when K30 1 % sensor self-calibration was used, highlighting the self-calibration capabilities of the
K-series sensors that contribute to their stable performance and high measurement repeatability with minimal maintenance
compared to other NDIR sensors.
**3.2 Field validation**
A total of 41 closed chamber measurements ($R_{eco}$: 21; NEE: 20) has been conducted during the two days field validation, using
the LI-850 as reference for both NDIR sensors passing the laboratory validation ($CO_2$; K30 FR and SCD30) and the two tested
RH sensors (ET; SHT31 and DHT22). While for the LI-850, 41 valid $CO_2$ fluxes ($R_{eco}$: 21; NEE: 20) could be calculated, 35
($R_{eco}$: 21; NEE: 14) and 36 ($R_{eco}$: 21; NEE: 15) valid fluxes were obtained for K30 FR and SCD30, respectively. To avoid
systematic impact of opaque chambers on plant transpiration via stomatal closure upon darkening, in case of ET fluxes, only
transparent chamber measurements were taken into account (Larcher, 2003). Out of the 20 NEE measurements, 13 valid ET
fluxes could be calculated in case of the LI-850. Compared to that, 18 and 17 valid ET fluxes were obtained for the SHT31
and DHT22, respectively. Differences in accuracy and precision for $CO_2$ and ET fluxes calculated based on NDIR (Fig. 6a-b)
and RH measurements (Fig. 6c-d) compared to $CO_2$ and ET fluxes calculated based on LI-850 are shown as 1:1-agreement
plots in Fig. 6. While the comparison between $R_{eco}$ and NEE fluxes calculated from LI-850 and K30 FR measurements (Fig.
6a), was in accordance with the laboratory validation and showed again the overall accuracy and precision of this NDIR sensor,
a small positive offset was found. Hence, $CO_2$ fluxes for the K30 FR were significantly higher ($R_{eco}$ mean diff. 1.12 µmol m$^{-2}$
s$^{-1}$; one sample t-test, p<0.05) and less negative (NEE mean diff. 1.41 µmol m$^{-2}$ s$^{-1}$; one sample t-test, p<0.05) when compared
to LI-850. No such systematic offset was found in case of the SCD30 (Fig. 6b), which showed significantly lower $R_{eco}$ (mean
diff. -1.33 µmol m$^{-2}$ s$^{-1}$; one sample t-test, p<0.05) and much less negative NEE fluxes (mean diff. -4.18 µmol m$^{-2}$ s$^{-1}$; one
sample t-test, p<0.05) compared to LI-850. Since neither both NDIR sensors showed a similar offset, nor an overestimation




was found for the K30 FR during the laboratory validation already, it can be assumed that the detected offset in case of the
K30 FR is neither a direct result of microclimatic effects (e.g., increasing humidity), nor incorrect sensor readings. Instead,
inter-alia differences within the chamber headspace and the position of the NDIR sensor right below the chamber top, approx.
10 cm above the LI-850 inlet and outlet, might help to explain it. Compared to the K30 FR, especially NEE fluxes obtained
by the SCD30, were characterized by a very low precision.

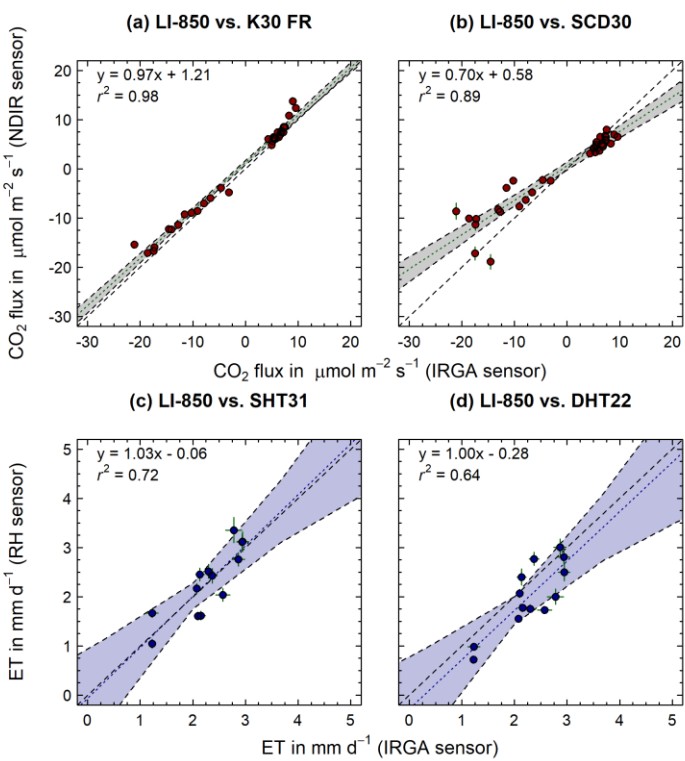


**Figure 6:** 1:1-agreement between (a-b) $CO_2$ and (c-d) ET fluxes measured with infrared gas analyzer (IRGA; LI-850, LI-COR,
USA) and low-cost NDIR sensors (K30 FR and SCD30). The dashed black line indicates the 1:1-agreement. The dotted
green/blue line shows the linear regression through the measured $CO_2$/ET fluxes. The grey/blue shaded area represents the
respective confidence band of the regression line. Error bars indicate calculated flux error (α=0.9).
The reason for this are certainly the lower $CO_2$ concentrations (<400 ppm) in the NEE measurements, which are clearly outside
the measurement range specified by the manufacturer (400 to 10000 ppm). This also explains the decreasing precision with
increased negative NEE fluxes obtained by SCD30, since these are likely related to $CO_2$ concentration measurements well



below 400 ppm. The general underestimation of $R_{eco}$ and NEE fluxes derived from SCD30, however, is probably a result of its
rather long response time and lower response strength when compared to the K30 FR (see 3.1). No significant difference (mean
diff. -0.01 mm d$^{-1}$; one sample t-test, p=0.89) was found between ET fluxes calculated from $H_2O$ concentration and RH
measurements, using the LI-850 and SHT31, respectively (Fig. 6c). Together with an r$^2$ of 0.72, this indicates a reasonable
accuracy of SHT31 derived ET flux estimates. Compared to that, ET fluxes, determined through RH measurements using the
DHT22 (Fig. 6d), were significantly smaller (mean diff. 0.28 mm d$^{-1}$; one sample t-test, p<0.05) than LI-850 based ET fluxes
and with an r$^2$ of 0.64, less accurate. This is consistent with sensor accuracy for measuring relative humidity specified by their
corresponding manufacturers, which are ±2 % accuracy for SHT31 and ±2-5 % accuracy for DHT22. Since these low-cost
sensors were only capable of measuring at this level of accuracy, a higher uncertainty at lower RH concentrations and
consequently derived ET fluxes, might occur, even though not directly detected within this study. The overall precision of
SHT31 and DHT22 derived ET fluxes were fairly similar, but with a residual SD of 0.36 and 0.39 mm d$^{-1}$, rather high. Figure
7 shows $R_{eco}$ (Fig. 7a-b) and GPP (Fig. 7c-d) parameter estimates for flux measurements performed with the LI-850 compared
to K30 FR (Fig. 7a, 7c) and SCD30 (Fig.7b, 7d), respectively. Since the $R_{eco}$ and GPP parameters are based on the fluxes
presented in Fig. 6, similar differences between LI-850, K30 FR and SCD30 could be obtained. With an $R_{ref}$ and $E_0$ of 4.60
and 212.71, the K30 FR had similar, but slightly higher $R_{eco}$ parameters (Fig. 7a) when compared to the LI-850 ($R_{ref}$: 4.14; $E_0$:
195.01). This indicates not only in general higher $R_{eco}$ fluxes but, more importantly, also a stronger increase of $R_{eco}$ fluxes with
rising temperature. In the case of the SCD30 ($R_{ref}$: 2.54; $E_0$: 270.07), differences in $R_{eco}$ parameters were, however, much more
pronounced. The same tends to be true for obtained GPP parameters, which were highly comparable for LI-850 (α: -0.048;
$GP_{max}$: -39.83) and K30 FR (α: -0.042; $GP_{max}$: -38.42), but distinctly different for SCD30 (α: -0.029; $GP_{max}$: -31.83). As a
result, the fitted K30 FR PAR dependency function was fully within the confidence band of the LI-850 PAR dependency
function. In summary, the K30 FR well represented $R_{eco}$ and GPP fluxes measured with the LI-850 and thereon based parameter
estimates for $R_{eco}$ and GPP. Unlike the K30 FR, the SCD30 was only able to reflect LI-850 $R_{eco}$ and GPP fluxes measured
within the manufacture specified concentration range. Correspondingly, accurate parameter estimates, especially with GPP,
were not obtained. Our findings are further supported by studies that compared the accuracy of K-series sensors against
commercial sensor counterparts and its accuracy for field $CO_2$ flux measurements (Curcoll et al., 2022). They integrated a K30
STA sensor into NFT-NSS chamber measurements and were able to accurately measure $CO_2$ fluxes for a grassland ecosystem.
Adding to that, the average $CO_2$ flux obtained during our study using K30 FR (0.4 µmol m$^{-2}$ s$^{-1}$) falls within the range of
reported daily average NEE values (4 to -6 µmol m$^{-2}$ s$^{-1}$) in the study by Emmel et al. (2018) for a field site in Switzerland
which was also covered with *Phacelia* cover crop. Based on the performed field validation, the developed low-cost
measurement device equipped with the K30 FR and SHT31 is likely to accurately measure $CO_2$ and ET fluxes in situ, using
NFT-NSS closed chambers.

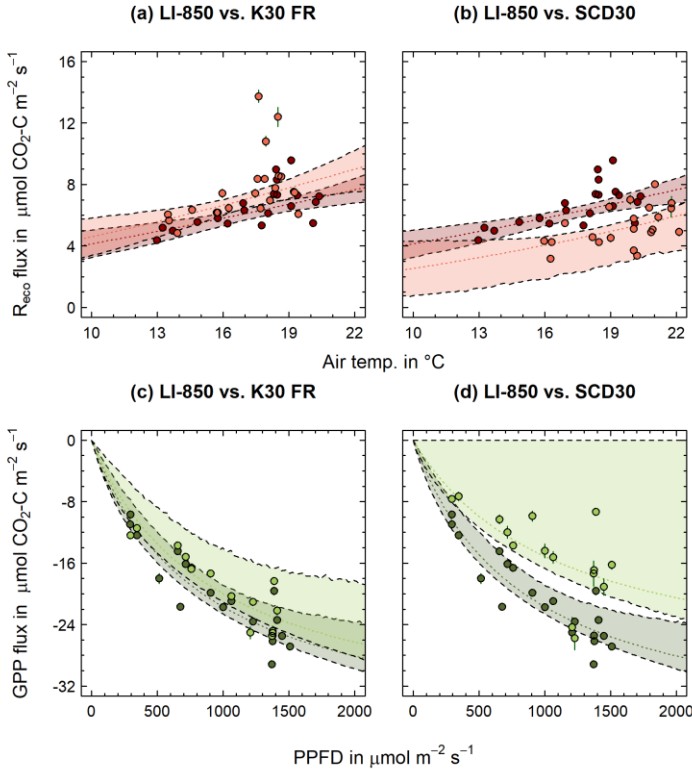

**Figure 7:** Comparison of $R_{eco}$ temperature dependency (dotted red lines) and GPP PAR dependency functions (dotted green lines) between LI-850 (dark red/green) and K30 FR and SCD30 (light red/green), respectively. Shaded red/green areas indicate confidence band around functions. Dots represent measured $R_{eco}$ and derived GPP fluxes. Error bars indicate calculated flux error ($\alpha=0.9$).

## 3.3 Field trial application

During the measurement period, half-hourly air temperatures at the field site near Nyankpala, Northern Ghana, reached as high as 46°C, with daily average air temperatures ranging from 24°C to 32°C. Daily rainfall varied strongly between the rainy and dry season, with single heavy rain event of up to 115 mm d$^{-1}$. Consequently, average monthly air humidity was highest (65 to 85 %) during the rainy season and as low as 23 % during the dry season. Irrespective of these harsh environmental conditions, the reliability of the developed low-cost measurement device could be proven during the field trial application. Periodically performed diurnal $CO_2$ measurement campaigns resulted in consistent $R_{eco}$ and NEE fluxes, showing throughout the entire crop growth a clear light (PAR) dependency for derived GPP fluxes (data not shown). The maximum daily $R_{eco}$ (3.9 g C m$^{-2}$





d$^{-1}$) and GPP (-6.9 g C m$^{-2}$ d$^{-1}$ ) fluxes derived for the non-mineral fertilized treatments, were well within the range (4.0 g C m$^{-}$
$^2$ d$^{-1}$ and -7.0 g C m$^{-2}$ d$^{-1}$) of EC derived maximum daily $R_{eco}$ and GPP fluxes reported by Quansah et al. (2015), who measured
a mixed fallow and cropping system in Northern Ghana, dominated by tall grasses. When adjusted for observation length,
cumulative NEE, GPP and $R_{eco}$ values obtained during the same study (27 g C m$^{-2}$, -195 g C m$^{-2}$ and 222 g C m$^{-2}$) were found
to be consistent with the average cumulative NEE, GPP and $R_{eco}$ values obtained from the non-mineral fertilized treatments
during our field trial application experiment (-58 g C m$^{-2}$, -355 g C m$^{-2}$ and 297 g C m$^{-2}$). Also, EC measurements of an
unfertilized cropland system (including maize) in Cameroon resulted with 218.5 g C m$^{-2}$ in a comparable cumulative $R_{eco}$
(Verchot et al., 2020). Regarding ET, the highest cumulative ET of our study (FM + MIN; 229 mm) was similar to the measured
ET flux (238 mm) of a field site in Northern Benin, which was dominated by C4 plants (Mamadou et al., 2016). In general,
obtained cumulative ET (Fig. 8d) for all four treatments were furthermore in a good agreement with ET obtained for Northern
Ghana from average monthly actual evapotranspiration (FAO, 2019), corrected using phenology specific crop factors for grain
maize (263 mm; Brouwer and Heibloem, 1986). Cumulative $R_{eco}$ and GPP fluxes recorded for the four different treatments
well-reflected the difference in harvested biomass (529 g C m$^{-2}$ for FT+MIN and 267 g C m$^{-2}$ for FM+MIN), with higher
cumulative $R_{eco}$ and GPP for higher crop biomass (Fig. 8a-b). Consequently, also NEE and thereon based NECB was higher
for additionally, mineral fertilized treatments compared to non-mineral fertilized treatments, with differences between
additionally, mineral and non-mineral fertilized treatments being more pronounced for FM when compared to FT (Fig. 8c and
e). Similar tendencies were found for ET and thereon based WUE, with additionally, mineral fertilized treatments showing a
higher ET and WUE compared to non-mineral fertilized treatments (Fig. 8d and f). This is in alignment with results reported
by Mo et al. (2017) for maize in Kenya, where WUE increased with higher grain yield due to increasing mineral N fertilization.
Besides the reliability of the developed low-cost measurement system, also its practicability was proved during the field trial
application. Despite of the rather demanding environmental conditions, the system showed that it is uncomplicated and easy
to operate even for untrained staff. It easily connects to end user devices using the Bluetooth module, so data can be visualized
inter-alia with a smartphone in real-time without the need to open the weather and shock resistant outdoor housing. After a
short training session, even non-technical trained staff can conduct minor repairs of the system directly in the field. Its light-
weight and low power consumption with the 12 rechargeable NiMH batteries lasting for as long as eight hours, make the
system especially suitable for in situ closed chamber measurements in remote tropical areas. Compared to Li-ion batteries, the
rechargeable NiMH batteries are furthermore relatively safe to use at high temperatures. However, the missing user interface
currently still prevents direct input of information, such as names of measurement location and soil temperatures, which made
data post processing more tedious.



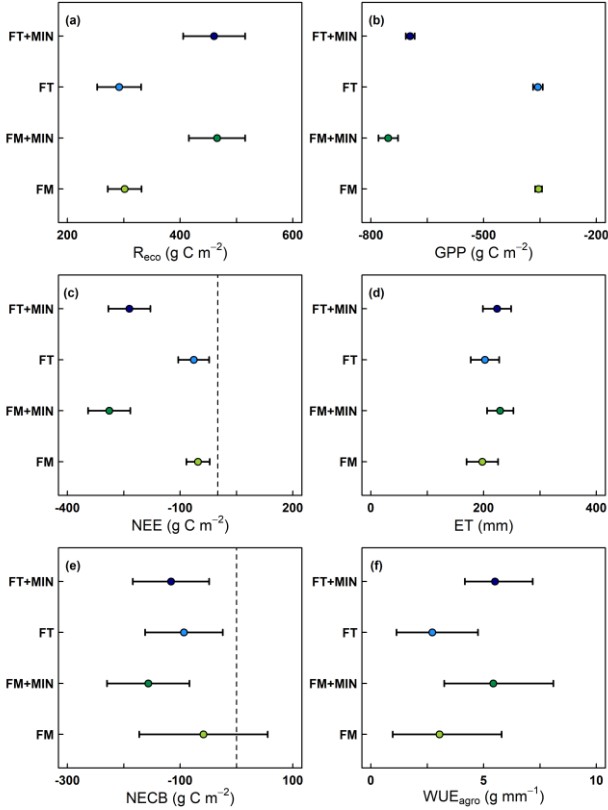


**Figure 8:** Cumulative (a-d) $R_{eco}$, GPP, NEE (g C m$^{-2}$) and ET fluxes (mm) as well as thereon based estimates of (e-f) NECB (g C m$^{-2}$) and WUE (g mm$^{-1}$) for the four different fertilizer treatments, namely: 1.) Fertisoil (5 t ha$^{-1}$; commercial organic fertilizer in Northern Ghana; FT), 2.) farmyard manure (5 t ha$^{-1}$; FM), 3.) Fertisoil + NPK (5 t ha$^{-1}$ + 90-60-60 kg ha$^{-1}$; FT+MIN) and 4.) farmyard manure + NPK (5 t ha$^{-1}$ + 90-60-60 kg ha$^{-1}$; FM+MIN).

## 4 Conclusions and implications for further use

Performed experiments showed that $CO_2$ and ET fluxes can be measured reliably and in a stable manner over time using inexpensive NDIR and RH sensors in conjunction with a manual closed chamber system. Out of the various low-cost $CO_2$ and RH sensors that were validated, the K30 FR and SHT31 proved to be the most accurate in measuring $CO_2$ and ET fluxes, respectively. Additionally, the developed low-cost measurement device was shown to be both practical and applicable to use even in environmentally challenging agroecosystems, as demonstrated by the field trial application in Northern Ghana, sub-Saharan Africa. There within, seasonal $CO_2$ and ET fluxes turned out to be reliable and could be used to obtain valid NECB



and WUE estimates. Since the system developed is battery-powered (solar rechargeable), based on open-source technology and all its components are low-cost, it can become easily accessible to a broad range of researchers. This opens manyfold potential applications, especially in the Global South, regarding the evaluation and identification of various land use systems and management practices, in terms of their C sequestration potential, water consumption and WUE. Therefore, the developed measurement device can be a valuable tool in evaluating and assessing global C and water flux models, ultimately expanding the network for C budget and ET research that are both critical for climate crisis adaptation and mitigation.

**5 Data and code availability**

The data and code referred to in this study are publicly accessible at https://doi.org/10.4228/zalf-hdqh-br42.

**6 Author contribution**

MH and RM conceptualized and developed the system and code. RM, DA and GS carried out the laboratory and field validation experiments. MA conducted the field trial application. RM, MH, and MD wrote and prepared the manuscript with contributions of all co-authors. All authors have reviewed and agreed to the final version of the manuscript.

**7 Competing interests**

The authors declare that they have no conflict of interest.

**8 Acknowledgements:**
This work was funded by a grant (2819DOKA06) from the German Federal Ministry of Food and Agriculture (BMEL). Michael Asante and Geoffroy Sossa were supported by the West African Science Service Center on Climate Change and Adapted Land Use (WASCAL) and the Prince Albert II of Monaco Foundation. We would also like to extend our deepest gratitude to Matthias Lueck and Shukrona Giyasidinova for assisting during the laboratory validation experiment, as well as Shrijana Vaidya and Isabel Zentgraf for helping during the field validation experiment. Special thanks goes to Ayertey Aquinas Kofi, Mavis Nartey, Narkey Kofi Mark and Abdul-Fataw Alhassan who helped during the field trial application.

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
