# Peer review of "Validation and field application of a low-cost device to measure CO2"

_EGUsphere, 2023_

## Author Comment (AC1)

We express our gratitude to the editor and the anonymous reviewers for their valuable feedback and for giving us the chance to revise and thereby improve our manuscript entitled: "Validation and field application of a low-cost device to measure $CO_2$ and ET fluxes." We have thoroughly addressed each comment from the reviewers in a detailed manner. Please take note of the color coding in our responses: (I.) reviewer comments are displayed in black; (II.) our responses are indicated in green; (III.) parts of the manuscript containing modifications are presented in italic and grey.

**General comments:**

This paper from Reena et al. does a good job by testing the use of low-cost devices to measure $CO_2$ and ET fluxes in agricultural soils in order to estimate, NEE, GPP, NECB and WAE. Not only the application is interesting but also the approach they have used, with a preliminary laboratory test, a field validation and finally the field trial application.

Nevertheless, I would suggest the authors to modify some parts of the manuscripts in order to make it clearer and more robust.

-In section 2.5.3. the author says that Error calculation for $CO_2$ and ET fluxes were quantified using a comprehensive error prediction algorithm described in detail by Hoffmann et al (2015). However, the reader would appreciate an understandable error analysis. For example, in figures 6 and 7 you say "error bars indicate calculated flux error ($\alpha$ =0.9). What does this $\alpha$ means? The only $\alpha$ I know in statistics is the *level of significance*, and is never higher than 0.05.

We now added more information on the used error prediction algorithm described in detail by Hoffmann et al. (2015) by adding the following to section 2.5.3:

*Error calculation for $CO_2$ fluxes, as well as crop season $CO_2$ exchange, were quantified using a comprehensive error prediction algorithm described in detail by Hoffmann et al. (2015). The approach utilizes bootstrapping alongside k-fold subsampling to estimate uncertainties for each flux measurement as*

*well as subsequent R_{eco} and GPP parametrization and final gap-filling. An adaptation of this approach was used to calculate errors in ET fluxes (Dahlmann et al., 2023). Seasonal ET flux errors were then estimated based on 1.96×SD of daily average ET fluxes.*

We appreciate the valuable comment on figures 6 and 7. We indeed accidentally mixed α with the confidence interval (CI) aimed to be reported here. To furthermore make the error analysis more understandable, we changed the reporting format in figure 6 and 7 captions, now including the CI and *p*-value as follows: *(CI: 95%; p<0.05).*

To do so we recalculated and applied in the corresponding figures the given errors for a 95% CI instead of the initially used 90% CI given in figure 6 and 7. However, we also want to emphasize that depending on the discipline and kind of measurements, the level of significance can be indeed higher than α=0.05 (e.g α=0.1) (Bonnet et al., 2021; Prado-Lorenzo et al., 2009; Friedl and Getzner, 2003)

Moreover, when you compared the fluxes from K30-SCD-30 with the Li-850 ones, you are talking about the r^2 (linearity) but it will be also interesting to know something about the error (RMSE, RSE, ...). Also, has the uncertainty of the measurement been taken in consideration when calculating the error of the fluxes?

We now added information about RMSE, RSE and MAE to section 3.2.2 to compare not only fluxes from K30FR and SCD30 with the LI-850 but also give further information on their error, as follows:

*Nonetheless, the NDIR sensor K30 FR still exhibited higher accuracy than the SCD30 when validated against LI-850 flux measurements. The root mean squared error (RMSE), mean squared error (MSE), and mean absolute error (MAE) obtained from the K30 FR (RMSE: 1.77 μmol m$^{-2}$ s$^{-1}$; MSE: 3.16 μmol m$^{-2}$ s$^{-1}$; MAE: 1.34 μmol m$^{-2}$ s$^{-1}$) were lower in comparison to SCD30 (RMSE: 3.97 μmol m$^{-2}$ s$^{-1}$; MSE: 15.77 μmol m$^{-2}$ s$^{-1}$; MAE: 2.80 μmol m$^{-2}$ s$^{-1}$).*

Due to the used bootstrapping approach, applied to concentration readings of each flux measurement, the measurement uncertainty, as any uncertainty within the measurement process (e.g. outliers, etc.) apart from

the actual sensor error (which was indirectly evaluated by comparison in lab and field validation) was accounted for when calculating the reported flux error.

Finally, in section 3.3 you enumerate all the parameters without citing any kind of deviation, error, variability or confidence interval. Please revise. Also indicate what the error bars means in figure 7.

We now added calculated errors to all parameters in section 3.3 obtained during our field trial application. In addition, error bars in all figures are now indicated in corresponding figure captions including figure 7. While also checking all given parameters thoroughly once more, we furthermore corrected cumulative biomass FM+MIN from the accidentally given average to the sum as mentioned in the section.

-In your manuscript you say that the LI-850 $CO_2$ values were corrected for $H_2O$. But you did not correct the K30 $CO_2$ values for $H_2O$ or Temperature. As you restricted the temperature increase to 1.5 ºK, maybe the temperature increase won't affect so much the readings. However, what about the influence of $H_2O$ increase? In some of the literature you cited (C.R. Martin et al., Curcoll et al. and others) there is an evaluation of how T and RH influences the measurement. Therefore, you can:

- recalculate your measurements applying an average correction for (T/$H_2O$) taken from the literature
- Estimate the error for not applying any correction in order to justify why you are not taking it in consideration.

We estimated the error for not applying a water correction (as temperature increased was restricted, no temperature correction was performed for neither LI-850 nor K30FR based $CO_2$ concentration readings; average temperature increase for fluxes identified for LI-850 and K30FR were 0.35°C and 0.71°C, respectively, both below 1°C). We recalculated K30FR $CO_2$ fluxes, correcting $CO_2$ concentration readings of the K30FR NDIR sensor with $H_2O$ measurements of the LI-850 (as reference) and SHT31, respectively. Recalculated $CO_2$ fluxes when using the LI-850 for $H_2O$ correction, differed in average by 0.5% from $CO_2$ fluxes without water correction. When using the SHT31, recalculated $CO_2$ fluxes differed in average by 1.2%. Hence, the error of not applying a water correction is lower than when applying a water correction using a low-cost RH sensor (taking the LI-850 as reference). Therefore, we decided to not apply a water

correction. In addition, e.g., during field validation, we aimed at comparing $CO_2$ fluxes calculated with measurements by the LI-850 (IRGA) and K30FR (NDIR) independent from potential biases introduced by the less accurate (see figure 6) low-cost RH sensors. However, it's important to note that even when using the low-cost SHT31 sensor for a water correction of the K30FR, the average error per flux was still $< 0.1$ $\mu$mol m$^{-2}$ s$^{-1}$.

**Specific comments:**

Figure 1c: which are the different elements represented the figure? They should be indicated (e.g. K30 sensor, Arduino board, etc…)

Different elements represented in Figure 1c were now indicated accordingly.

Paragraph beginning in line 117: please re-write in order to make it easier to read and understand. Make it shorter and enumerate the sensors at the end of the phrase.

Changed as follows:

*To identify the NDIR sensor most suitable for in situ, dynamic closed chamber measurements, four different NDIR-based sensors were tested and validated regarding their precision and accuracy during a laboratory validation experiment. The sensors tested were 1.) MH-Z19 (Winsen Electronics Technology CO., LTD, China), 2.) MH-Z14 (Winsen Electronics Technology CO., LTD, China), 3.) SCD30 (Sensirion AG, Switzerland) and 4.) K30 FR (Senseair AB, Sweden).*

Line 160: What do you mean for "*changes in the chamber headspace*"?

"Changes in Chamber headspace" refers to changes of $CO_2$ and $H_2O$ concentration in the chamber headspace during chamber closure. To better express this, the sentence was revised as follows:

*During individual 4 min measurements, $CO_2$ and $H_2O$ concentration changes in the chamber headspace, as well as RH, air temperature inside and outside the chamber, soil temperature and humidity (TMS-4,*

*TOMST, Czech Republic) as well as PAR (outside the chamber; Skye, UK) were recorded at a 3 s (LI-850) and 5 s interval (NDIR and RH sensors).*

Line 163: "*derived temperature (Reco)*". Is this correct?

$R_{eco}$ written in brackets refers to the temperature dependency function coming later within this sentence. We revised the sentence to avoid any misunderstanding as follows: *To validate the low-cost $CO_2$ and ET flux measurement device, measured $R_{eco}$, NEE, and ET fluxes, as well as the derived temperature and PAR dependency functions for $R_{eco}$ and GPP, respectively, were directly compared with results obtained in parallel with the LI-850.*

Line 196: "*death band of 10%*". What do you mean for a "death band"? Correct or specify.

Death band of 10% is a user defined setting that was specified within the modular R script by Hoffmann et al. (2015) used for $CO_2$ flux calculation. This was performed such that first and last 10% of each measurement were removed prior to flux calculation to exclude data noise from turbulences and pressure fluctuation caused by chamber deployment (Hoffmann et al., 2015), and to avoid biases from time required to homogenize chamber headspace air (Vaidya et al., 2021). To avoid confusion we changed the term to trimming, which is more common in statistics and data processing. In addition a short explanation of what it does and what it purpose is was added to the MS as follows:

*Prior to $CO_2$ and ET flux calculation, underlying data was trimmed by removing the first and last 10 % of each chamber measurement dataset. This was conducted to eliminate data noise caused by turbulences and pressure fluctuations due to chamber deployment (Hoffmann et al., 2015), and to mitigate biases arising from the time needed to homogenize chamber headspace air (Vaidya et al., 2021).*

Line 222: ”*steepest slope and closest to chamber deployment*”. Why the steepest slope? Justify, with reference. What you meant by "closest to chamber deployment"??

The "steepest slope" was taken as a final choosing criteria after all other criteria were met, hence it does not mean that in general the steepest slope of each measurement was taken for subsequent flux calculation. We chose the steepest or highest regression slope, in order to prevent potential underestimation of measured flux rates caused by saturation within the chamber headspace. Preference towards higher regression slopes is consistent with numerous studies by Vaidya et al. (2021), Pirk et al. (2016), Bäckstrand et al. (2008), and Bubier et al. (2003).

For "closest to chamber deployment", we meant choosing measurements closest to the point of chamber closure or when the chamber was securely placed on the frame (Rochette and Hutchinson, 2015). We revised the sentence to avoid any misunderstanding as follows:

*In cases where more than one flux per measurement met all criteria, the $CO_2$ and ET flux with steepest slope and closest in time to chamber closure were chosen.*

Figure 6: As in the text you talk about Reco and NEE fluxes, you could differentiate it by using different colours or point shapes.

$R_{eco}$ and NEE fluxes in Figure 6 are now differentiated using different colors.

[Figure]

*Figure 6: 1:1-agreement between (a-b) $CO_2$ ($R_{eco}$: dark red points; NEE: dark green points) and (c-d) ET fluxes measured with infrared gas analyzer (IRGA; LI-850, LI-COR, USA), and low-cost NDIR sensors (K30 FR and SCD30), as well as low-cost RH sensors (SHT 31 and DHT22), respectively. The dashed black line indicates the 1:1-agreement. The dotted green/blue line shows the linear regression through the measured $CO_2$/ET fluxes. The grey/blue shaded area represents the respective confidence band of the regression line. Error bars indicate calculated flux error (CI: 95%; $p<0.05$).*

Lines 415 to the end of the section: this paragraph may be split. Some of the information you write must go to methods section, and some other in the conclusions.

Following this suggestions, Lines 415 until the end of the corresponding section was changed. The following sentence was transferred to the method section (2.1 Hard- and software implementation):

*It easily connects to end user devices using the Bluetooth module, so data can be visualized inter-alia with a smartphone in real-time without the need to open the weather and shock resistant outdoor housing.*

In addition to that the following sentences were transferred to the conclusion:

*Since the system developed is battery-powered (solar rechargeable), based on open-source technology and all its components are low-cost, it can become easily accessible to a broad range of researchers. Its light-weight and low power consumption with the 12 rechargeable NiMH batteries lasting for as long as eight hours, make the system especially suitable for in situ closed chamber measurements in remote tropical areas.*

**References:**

Bäckstrand, K., Crill, P. M., Mastepanov, M., Christensen, T. R., and Bastviken, D.: Non-methane volatile organic compound flux from a subarctic mire in Northern Sweden, Tellus B: Chemical and Physical Meteorology, 60, 226, https://doi.org/10.1111/j.1600-0889.2007.00331.x, 2008.

Bonnet, R., Swingedouw, D., Gastineau, G., Boucher, O., Deshayes, J., Hourdin, F., Mignot, J., Servonnat, J., and Sima, A.: Increased risk of near term global warming due to a recent AMOC weakening, Nat Commun, 12, 6108, https://doi.org/10.1038/s41467-021-26370-0, 2021.

Bubier, J., Crill, P., Mosedale, A., Frolking, S., and Linder, E.: Peatland responses to varying interannual moisture conditions as measured by automatic $CO_2$ chambers, Global Biogeochemical Cycles, 17, 2002GB001946, https://doi.org/10.1029/2002GB001946, 2003.

Dahlmann, A., Hoffmann, M., Verch, G., Schmidt, M., Sommer, M., Augustin, J., and Dubbert, M.: Benefits of a robotic chamber system for determining evapotranspiration in an erosion-affected,

heterogeneous cropland, Hydrology and Earth System Sciences, 27, 3851–3873, https://doi.org/10.5194/hess-27-3851-2023, 2023.

Friedl, B. and Getzner, M.: Determinants of CO2 emissions in a small open economy, Ecological Economics, 45, 133–148, https://doi.org/10.1016/S0921-8009(03)00008-9, 2003.

Hoffmann, M., Jurisch, N., Albiac Borraz, E., Hagemann, U., Drösler, M., Sommer, M., Augustin, J.: Automated modeling of ecosystem CO2 fluxes based on periodic closed chamber measurements: a standardized conceptual and practical approach, Agric. For. Meteorol. 200, 30–45, https://doi.org/10.1016/j.agrformet.2014.09.005, 2015.

Pirk, N., Mastepanov, M., Parmentier, F.-J. W., Lund, M., Crill, P., and Christensen, T. R.: Calculations of automatic chamber flux measurements of methane and carbon dioxide using short time series of concentrations, Biogeosciences, 13, 903–912, https://doi.org/10.5194/bg-13-903-2016, 2016.

Prado-Lorenzo, J., Rodríguez-Domínguez, L., Gallego-Álvarez, I., and García-Sánchez, I.: Factors influencing the disclosure of greenhouse gas emissions in companies world-wide, Management Decision, 47, 1133–1157, https://doi.org/10.1108/00251740910978340, 2009.

Rochette, P. and Hutchinson, G. L.: Measurement of Soil Respiration in situ: Chamber Techniques, in: Agronomy Monographs, edited by: Hatfield, J. L. and Baker, J. M., American Society of Agronomy, Crop Science Society of America, and Soil Science Society of America, Madison, WI, USA, 247–286, https://doi.org/10.2134/agronmonogr47.c12, 2015.

Vaidya, S., Schmidt, M., Rakowski, P., Bonk, N., Verch, G., Augustin, J., Sommer, M., and Hoffmann, M.: A novel robotic chamber system allowing to accurately and precisely determining spatio-temporal CO2 flux dynamics of heterogeneous croplands, Agricultural and Forest Meteorology, 296, 108206, https://doi.org/10.1016/j.agrformet.2020.108206, 2021.

---

## Author Comment (AC2)

We express our gratitude to the editor and the anonymous reviewers for their valuable feedback and for giving us the chance to revise and thereby improve our manuscript entitled: "Validation and field application of a low-cost device to measure $CO_2$ and ET fluxes." We have thoroughly addressed each comment. For convenience we color coded our responses as follows: black (reviewer comments), green (response), gray/italic (changed text from MS).

The paper presents the laboratory testing for $CO_2$ flux low-cost sensors (NDIR sensors), and field validation and field application of the same sensors together with evapotranspiration (ET) flux low-cost sensors (Relative Humidity sensors).

The experiments and analyses are very thorough and the paper makes a significant contribution to the low-cost sensor literature for $CO_2$ and ET fluxes. The authors provide good discussion of their results based on recent literature.

My main comment, in addition to minor edits suggested below, is that the results are sometimes difficult to read, with findings about each category of sensor ($CO_2$ and ET) being mixed up, and important findings such as the temperature dependency are buried in the text.

I recommend that the authors use subsections and indicate in subtitles which sensors they related to (e.g. l. 351, this should be a separate section for RH sensors). This will avoid some possible confusion, e.g., there was no lab testing for the RH sensors, but it is not clear in how the results are presented.

We created the suggested subsections as follows:

*3.2 Field validation*

- *3.2.1. In situ ET flux validation*

- *3.2.2. In situ $CO_2$ flux validation*

- *3.2.3 Temperature- and PAR-dependency of measured $CO_2$ fluxes*

**Minor comments:**

Table 1: In a separate section of the table, please add cost of the other NDIR and RH sensors tested (those that were not ultimately used in the field); this information is useful for the emerging low-cost sensor body of literature

As suggested, we added a separate section in the table for the other additional NDIR $CO_2$ sensors tested as follows:

| COST OF OTHER NDIR SENSORS TESTED | | | | |
|---|---|---|---|---|
| SENSIRION SCD30 MODULE | 1 | NDIR gas sensor for $CO_2$ (0-10000 ppm) integrated with humidity and temperature sensor in the same module | 63.50 Euro | www.berrybase.de |
| MH-Z14 $CO_2$ SENSOR MODULE | 1 | NDIR gas sensor for accurately measuring the $CO_2$ concentration (0-10000 ppm) | 55.60 Euro | www.kaufland.de |
| MH-Z19 $CO_2$ SENSOR MODULE | 1 | NDIR gas sensor for accurately measuring the $CO_2$ concentration (0-10000 ppm) | 28.50 Euro | www.reichelt.de |

Figure 1c: Please add component names to improve readability.

Changed accordingly.

[Figure]

*Figure 1: (a) Logger unit in weather and shock resistant housing, (b) external sensor unit attached to a transparent non-flow-through non-steady-state (NFT-NSS) closed chamber and (c) schematic representation of wiring.*

Figure 6: Caption should mention SHT31 and DHT22 as RH low cost sensors

SHT31 and DHT22 are now named as RH low-cost sensors in the caption of Figure 6.

l.121: units should be $cm^3$

Changed accordingly.

l.214: how did the authors identify the starting point for the moving window analysis? or did they use multiple starting points and multiple windows (0.5 to 3 min)?

Indeed multiple moving windows were used (0.5 min to 4 min). The script described in detail by Hoffmann et al. (2015) uses a variable moving window to calculate all possible subsets of a flux measurements and subsequent uses exclusion and quality criteria to identify the final flux. To make this clearer, we added the following to L214:

*The variables T and, more importantly, Δc/Δt, were obtained by applying a variable (window size 0.5 to 4 min) moving window to each chamber measurement.*

l. 323: Why less $CO_2$ fluxes could be calculated for the low cost sensors?

Flux calculation of closed chamber measurements needed to pass the same rigorous flux calculation algorithm for all sensors, as described in section 2.5.1. In case of K30 FR and SCD30 especially NEE measurements did not yield in valid flux estimates and thus did not passed this step.

This might be e.g., due to non-significant regression slope, non-linear concentration increase, variance inhomogeneity, outliers or last but not least larger variations in temperature or especially PAR. Since NDIR sensors are passive sensors they have a higher delay time than the LI-850. This can result in a shift of measured PAR and adequate measurement subsets thus attributing a high PAR variation to proper measurements subsets and vice versa during conditions characterized by a persistent change between sunny and cloudy conditions. Hence, as commonly aimed at, measurements should be best performed during sunny conditions.